# Efficacy and Safety of Ferrous Bisglycinate and Folinic Acid in the Control of Iron Deficiency in Pregnant Women: A Randomized, Controlled Trial

**DOI:** 10.3390/nu14030452

**Published:** 2022-01-20

**Authors:** Akkarach Bumrungpert, Patcharanee Pavadhgul, Theera Piromsawasdi, M. R. Mozafari

**Affiliations:** 1Research Center of Nutraceuticals and Natural Products for Health & Anti-Aging, College of Integrative Medicine, Dhurakij Pundit University, Bangkok 10210, Thailand; 2Department of Nutrition, Faculty of Public Health, Mahidol University, Bangkok 10400, Thailand; patcharanee.pav@mahidol.ac.th; 3Health Promotion Hospital, Health Center Region 5, Ratchaburi 70000, Thailand; drtheera@gmail.com; 4Australasian Nanoscience and Nanotechnology Initiative (ANNI), Monash University LPO, Clayton, VIC 3168, Australia; dr.m.r.mozafari@gmail.com

**Keywords:** iron deficiency, pregnant women, ferrous bisglycinate, folinic acid, absorption, iron status, side effects

## Abstract

Iron deficiency in pregnancy is a major public health problem that causes maternal complications. The objective of this randomized, controlled trial was to examine the bioavailability, efficacy, and safety of oral ferrous bisglycinate plus folinic acid supplementation in pregnant women with iron deficiency. Subjects (12–16 weeks of gestation, *n* = 120) were randomly allocated to receive oral iron as ferrous bisglycinate (equiv. iron 24 mg) in supplement form with folinic acid and multivitamins (test group, *n* = 60) or as ferrous fumarate (equiv. iron 66 mg iron, control group, *n* = 60) after breakfast daily. Iron absorption was assessed by measuring fasted serum iron levels at 1 and 2 h immediately after supplementation. Hematological biomarkers and iron status were assessed before intervention, and at 3 and 6 months. Side effects were monitored throughout the intervention. A significant increase in serum iron was seen in both groups (*p* < 0.001) during the bioavailability assessment; however, the test group increases were comparatively higher than the control values at each timepoint (*p* < 0.001). Similarly, both test and control groups demonstrated a statistically significant increases in hemoglobin (Hb) (*p* < 0.001), erythrocytes (*p* < 0.001), reticulocytes (*p* < 0.001), mean corpuscular volume (MCV) (*p* < 0.001), mean corpuscular hemoglobin (MCH) (*p* < 0.001), mean corpuscular hemoglobin concentration (MCHC) (*p* < 0.001), % transferrin saturation (*p* < 0.001), and ferritin (*p* < 0.001) at 3 and 6 months after supplementation. However, in all cases, the test group increases were numerically larger than the control group increases at each timepoint. The test intervention was also associated with significantly fewer reports of nausea, abdominal pain, bloating, constipation, or metallic taste (*p* < 0.001). In conclusion, ferrous bisglycinate with folinic acid as a multivitamin nutraceutical format is comparable to standard ferrous fumarate for the clinical management of iron deficiency during pregnancy, with comparatively better absorption, tolerability, and efficacy and with a lower elemental iron dosage.

## 1. Introduction

Iron deficiency during pregnancy is a preventable cause of several health problems in both the mother and infant, including an increased risk of anemia, premature birth, low birth-weight, developmental abnormalities, and postpartum depression [1,2]. Physiologically, pregnant women have a higher iron demand, which starts shortly after conception and increases gradually during gestation [3]. Iron is essential to support the increases in maternal erythropoiesis and to meet the requirements of the fetal organogenesis, notably the development of the central nervous system and hematopoietic tissues [4]. However, iron deficiency continues to be one of the most prevailing single-nutrient deficiencies worldwide, especially in third-world countries. Clinical and public health strategies for women and children have therefore involved oral supplementation with mineral iron forms to counteract the depletion of iron stores during pregnancy and lactation. An alternative public health approach has been the fortification of food items with purified iron sources. Throughout these interventions, the most common challenges with oral supplementation have arisen from the significant variability in the bioavailability and oral tolerability of different iron forms, especially conventional mineral forms, such as ferrous sulfate [5]. Pregnant women are particularly prone to the gastrointestinal side effects of these forms. Alternative forms have included iron chelates. One in particular, ferrous bisglycinate, has been demonstrated to have at least two-fold higher bioavailability and absorption compared to conventional iron salts, including ferrous sulfate and ferrous fumarate, while also resulting in improved oral tolerability during pregnancy [6,7]. It is well accepted that the main adverse effects of mineral iron are dose dependent. Consequently, utilization of iron sources with lower effective doses and improved compliance could alleviate the problems caused by inorganic iron and, most importantly, may prevent iron overload.

Iron absorption and metabolism is also dependent on other micronutrients, including the B-group vitamins. Maternal folate is a key correlate of hematological status, and as a result, iron and folic acid are routinely co-supplemented. Folinic acid, a more readily absorbed form of folic acid, is able to be converted more directly to tetrahydrofolate via the folic acid cycle. A recent study has shown that folinic acid in combination with chelated iron has proven to be more effective than iron alone for iron-deficiency anemia during pregnancy [8]. In light of this evidence, the objective of the present study was to evaluate the feasibility of a nutraceutical formulation containing ferrous bisglycinate and folinic acid in improving hematological and iron status in pregnant women with iron deficiency.

## 2. Materials and Methods

### 2.1. Subjects and Study Design

This clinical study was approved by the Ethics Review Committee for Human Research, Faculty of Public Health, Mahidol University (Application No. MUPH 2018-103), and conducted in accordance with the Declaration of Helsinki regulating research on human subjects. The trial protocol was registered with the Thai Clinical Trials Registry (TCTR20171207001) and the WHO International Clinical Trials Registry Platform (WHO-ICTRP).

A sample size of 120 was considered as adequately powered, based on discriminating a 10% difference in iron status as the primary endpoint indicator of treatment versus control, with a 10% standard deviation (SD) of effect (α = 0.05 and β − 1 = 0.8) and an anticipated dropout rate of 10%. A total of 324 pregnant women were recruited and screened at the Antenatal Care Clinic of the Health Promotion Hospital, Health Center Region 5, Ratchaburi, Thailand, from October 2018 to June 2019. Exclusion criteria included gestational diabetes mellitus; pre-eclampsia; thalassemia; haemoglobinopathies; chronic diarrhea or other gastrointestinal disorders; treatment with aspirin; and smoking and alcoholism. After screening, a total of 120 pregnant women (20–40 years, 12–16 weeks of gestation) with iron deficiency (serum ferritin < 30 µg/L) were included, with all women giving informed consent and indicating willingness to follow trial procedures prior to enrollment.

The study was designed as a simple randomized, controlled trial (Figure 1), with participants randomly allocated to a treatment or a control group based by an independent researcher using a computer-generated sequence (www.randomization.com, accessed on 24 June 2019). The treatment group (*n* = 60) was given a nutraceutical supplement capsule containing 120 mg ferrous bisglycinate (24 mg of elemental iron) plus folinic acid and multivitamin (IronUp^®^, Max Biocare Pty Ltd., South Yarra, Victoria, Australia, composition shown in Table 1). The control group (*n* = 60) was given a supplement containing 200 mg ferrous fumarate (66 mg of elemental iron) plus multivitamin (The Government Pharmaceutical Organization (GPO) Thailand, composition shown in Table 1) after breakfast daily for 6 months. Clinical examinations were conducted by a physician and a research assistant. Subjects kept daily study records, which were reviewed by the research assistant on a weekly basis.

Primary outcomes included vital signs (blood pressure, weight, body mass index, and pulse) and blood chemistry (hematological and iron status and inflammatory status (high sensitivity C-reactive protein) as detailed below. Secondary outcomes included fasted iron absorption, frequency of side effects, quality of life, and newborn weight. We assessed the frequency of primarily gastrointestinal side effects, including nausea, vomiting, abdominal pain, bloating, constipation, diarrhea, and metallic taste, using a participant questionnaire adapted from a previous study [9], Maternal Quality of Life (QOL) index. The latter was assessed using the World Health Organization Quality of Life Brief survey in Thai (WHOQOL-BREF-THAI) as reported previously [10].

### 2.2. Biochemical Analysis

Blood samples were collected by a registered nurse at baseline, 3-, and 6-month timepoints for the determination of serum hemoglobin (Hb) concentration, erythrocyte and reticulocyte counts, mean corpuscular volume (MCV), mean corpuscular hemoglobin (MCH), mean corpuscular hemoglobin concentration (MCHC), transferrin saturation percentage, ferritin concentration, and high sensitivity C-reactive protein (hs-CRP) concentration. Iron absorption was assessed by measurement of fasted serum iron levels from blood sampled at baseline, 1-, and 2-h timepoints. All blood samples were drawn from the antecubital vein and transported to N Health Asia Lab, Bangkok, Thailand, a global clinical testing laboratory for biochemical analysis.

### 2.3. Statistical Analysis

Statistical analysis was performed on all endpoint parameters in the per-protocol group (*n* = 108 who completed the trial). All parameter values across all timepoints were analyzed, and no data were excluded. Normality was analyzed using the Shapiro–Wilk test. The ROUT test was used to identify outliers, which were omitted at Q = 0.1%. For comparison between normally distributed values at specific timepoints, repeated measures ANOVA with Bonferroni correction was performed. The Kruskal–Wallis rank-sum test was performed for multiple comparisons of non-normally distributed data both between and within groups. An independent *t*-test was used to analyze the differences in newborn weight between the test group and control group. Friedman’s test was performed for comparison of matched data within groups without missing datapoints. The chi-square test was used to compare reported proportions of side effects between test and control groups at the end of the trial. All statistical analyses were performed using Graphpad Prism version 8.3.0 and considered biologically significant for *p*-values < 0.05.

## 3. Results

### 3.1. Characteristics of Subjects

General participant characteristics are presented in Table 2. Age, weight, BMI, blood pressure, pulse rate, and hs-CRP levels were not significantly different between the test and control groups before and after the intervention. Five subjects in the test group and seven subjects in the control group dropped out, leaving a total of 108 subjects who completed the study as per protocol (Figure 1). Compliance was high in both the test (97%) and control (92%) groups based on the evaluation of unused capsules.

### 3.2. Iron Absorption

There were no significant differences in serum iron between the ferrous bisglycinate plus folinic acid group and the control group at baseline. A significant within-group increase in serum iron (*p* < 0.001) was found in both groups at 1 h and 2 h after intake of their allocated intervention. However, the increase in serum iron in the test group was significantly more pronounced in comparison to the control group at each timepoint following product intake (*p* < 0.001) (Table 3).

### 3.3. Effects of Supplements on Biomarkers of Hematological and Iron Status

Hematological and iron status for test and control participants at all timepoints are presented in Table 4. Baseline levels did not differ significantly between the two groups. The test group showed a statistically significant increase in Hb (*p* < 0.001), erythrocytes (*p* < 0.001), reticulocytes (*p* < 0.001), MCV (*p* < 0.001), MCH (*p* < 0.001), MCHC (*p* < 0.001), and % transferrin saturation (*p* < 0.001) and ferritin (*p* < 0.001) compared with control group at both the three- and six-month timepoint following supplementation, with a more pronounced numerical increase generally observed in the ferrous bisglycinate plus folinic acid group.

### 3.4. Maternal Side Effects, Quality of Life, and Birth Outcome

A statistically significant difference between the ferrous bisglycinate plus folinic acid group and the control group was observed in the reported frequencies of nausea, abdominal pain, bloating, constipation, and metallic taste (*p* < 0.001). A borderline significant reduction in vomiting was also seen; however, there was no change in the incidence of diarrhea (Table 5). A small decrease in QOL index was reported within both test and control groups at three and six months after supplementation (*p* < 0.05). However, this trend in QOL-score across pregnancy was less pronounced in the test subjects versus control subjects at three months (*p* = 0.013) and six months (*p* = 0.004) (Table 6). At the end of pregnancy, a small but significant (*p* = 0.029) increase in newborn weight was observed in the iron bisglycinate plus folinic acid group compared to the control group (Table 7).

## 4. Discussion

In this clinical study, we demonstrate that iron supplementation as ferrous bisglycinate with folinic acid and multivitamins improves hematological parameters, iron absorption, quality of life, and birthweight in iron-deficient pregnant women compared to ferrous fumarate. Supplementation with ferrous bisglycinate plus folinic acid was associated with fewer side effects and comparable compliance to ferrous fumarate but at a lower elemental iron dose (24 mg vs. 66 mg). We also show that general health parameters were not significantly different between the test and control groups, including no change in the inflammatory marker, hs-CRP, which has previously been correlated with a lower Hb response following iron supplementation [11].

The improvements in iron absorption and tolerability were also in agreement with previous evidence [12]. Previous research has demonstrated the effective absorption of ferrous bisglycinate in celiac patients [13] and similar hematological outcomes with much lower supplement dosages in pregnant women [7,14,15]. In a pilot study by Szarfarc et al., 15 mg/day of ferrous bisglycinate was found to be comparable to ferrous sulfate in terms of maternal blood parameters, while iron depletion was significantly less frequent [7]. Ferrous bisglycinate has also proven clinically effective for preventing iron deficiency anemia (IDA) in infants [6]. This comparative study reported that while 5 mg/kg (equiv. 30–48 mg/day) of ferrous sulfate was equally as effective as a similar dose of ferrous bisglycinate for improving hemoglobin levels, the bisglycinate form was superior in terms of plasma ferritin levels and overall bioavailability. This confirms an earlier study by the same investigators showing that 30 mg iron as bisglycinate is as effective as 120 mg of ferrous sulfate in preventing IDA in adolescents [16]. A more recent clinical study on iron-deficient school-aged children showed that 30 mg of either iron sulfate or bisglycinate supplementation resulted in similar improvements; however, the bisglycinate form showed more persistent correction of ferritin levels after six-month follow-up [17]. A clinical trial by Milman et al. [14] has shown that 25 mg/day of ferrous bisglycinate resulted in similar maternal blood parameters to women who took ferrous sulfate (50 mg/day). Similarly, a more recent clinical trial in pregnant women with IDA demonstrated that ferrous bisglycinate was more efficacious in increasing Hb level than ferrous glycine sulfate. Importantly, in three of the studies outlined here, ferrous bisglycinate was associated with fewer gastrointestinal side effects and high rates of compliance during pregnancy [7,14,18].

The present intervention also includes micronutrients that play important roles in the support of absorption and metabolism of iron, hemoglobin, and erythrocytes. Folinic acid, which in some forms is registered for use as a drug, is a readily absorbed form of bioavailable folates (vitamin B9) and was first used effectively to treat megaloblastic anemia and pernicious (vitamin B12-deficiency) anemia [19,20]. A recent clinical trial has reported that folinic acid in combination with iron chelate supplementation was more effective than iron alone for IDA during pregnancy [8]. While both treatments resulted in increased hemoglobin levels, the increase was more marked with iron/folinic acid, especially for women with lower-than-normal hemoglobin. Folate status is especially relevant in pregnant women, who have an approximately three-fold higher folate demand and, if anemic, often suffer from folate deficiency [21]. Based on previous clinical studies, it has been argued that iron/folate supplement combinations be implemented routinely [22]. Today, the WHO continues to advocate the use of folates with iron for this risk group to manage the risk of maternal IDA, infant IDA, low birth weight, and other gestational complications, especially in developing countries with low protein consumption [23]. Several clinical trials reported the benefits of combined administration of folate, iron, and/or multiple micronutrients before and/or during pregnancy to prevent birth complications [24].

The role of ascorbic acid in iron homeostasis relates to its prolific ability to enhance enteric iron absorption. It is effective at restoring iron balance and imparts metabolic benefits, owing to its antioxidant and anti-inflammatory properties. Ascorbic acid alone over a two-month period has been definitively shown to correct IDA in supplemented subjects [25]. The cited study reported a significant post-treatment increase in blood levels of iron, hemoglobin, and ferritin. Using iron in combination with ascorbate, another comparative study showed an improvement in hemoglobin recovery after transient erythrocyte deficits following blood donation in healthy females [26]. Moreover, while ascorbate and iron are highly effective in improving ferritin saturation and hepcidin levels at lower doses, this improvement is blunted in overweight and obese women [27].

Other B-group vitamins, including B1, B2, B6, and B12, play critical metabolic roles in iron homeostasis. The WHO advocates the use of multi-micronutrient supplements in risk groups, such as pregnant women, especially in developing countries where there is a high prevalence of nutrient deficiencies. The B vitamins listed above are common to several supplement types that have been issued for this purpose [28]. A recent comparative study in mildly anemic pregnant women evaluated supplements containing iron and folate with or without multi-micronutrients, including B-group vitamins. Supplementation of multi-micronutrient resulted in a modest early improvement in baseline serum ferritin and iron levels compared to iron/folate alone and further improvement over folate alone [29]. Similar results were reported in a much larger trial of over 6000 pregnant women. Of the groups—(a) 400 μg folate alone, (b) 400 μg folate/40 mg iron, or (c) both groups with multi-micronutrient—regimens of (b) and (c) resulted in a significant reduction in the frequency of maternal anemia in late gestation compared to group (a), with no differences in fetal morbidity or mortality across all treatments [30]. As discussed earlier, a meta-analysis [24] of trials where multi-micronutrient combinations have been used with iron/folate show consistent benefits in preventing anemia during pregnancy.

The major strengths of the present study include its randomized, controlled design and the number of subjects, which represents a sample that was sufficiently powered to demonstrate significant effects. Importantly, the trial demonstrates that iron bisglycinate provides a safer and more amenable iron form because of its improved absorption, improved tolerability, and lower dose. Furthermore, the absence of subjects with chronic inflammatory indicators eliminated any confounding effects on iron metabolism. Some limitations of our study include firstly that no food record for dietary assessment of intake was made to control for levels of micronutrients found in the test product. Secondly, we did not measure folate or B-vitamin status, so we cannot conclude on how these vitamins may have contributed to the improvements observed in iron status. Thirdly, we did not control for any other variables that may have impacted on quality of life during pregnancy. Finally, there was no blinding in this study.

## 5. Conclusions

The present report demonstrates that ferrous bisglycinate plus folinic acid (24 mg elemental iron) supplementation can improve hematological and iron status in pregnant women with iron deficiency, with fewer side effects than a ferrous fumarate (66 mg elemental iron) preparation. Therefore, the commercial product may be considered feasible for clinical and nutritional strategies to manage iron deficiency during pregnancy, reaffirming the need for more efficient iron delivery and consideration of physiological aspects of iron homeostasis rather than simply titrating levels or frequencies.

## Figures and Tables

**Figure 1 nutrients-14-00452-f001:**
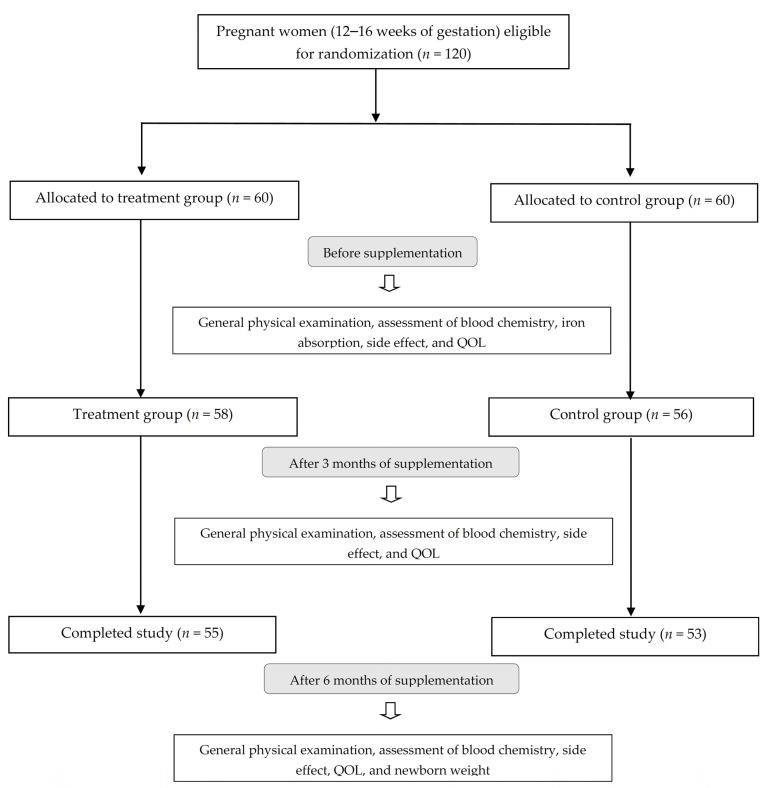
Flow chart of study participants.

**Table 1 nutrients-14-00452-t001:** Supplement intervention between both groups.

Ferrous Bisglycinate and Folinic Acid		Control	
Ferrous bisglycinate	120 mg	Ferrous fumarate	200 mg
Equiv. iron	24 mg	Equiv. iron	66 mg
Folic acid (vitamin B9)	400 mcg	Folic acid (vitamin B9)	400 mcg
Calcium folinate	127 mcg	Calcium folinate	-
Equiv. folinic acid	100 mcg	Equiv. folinic acid	-
Ascorbic acid (vitamin C)	50 mg	Ascorbic acid (vitamin C)	50 mg
Thiamine nitrate (vitamin B1)	5 mg	Thiamine nitrate (vitamin B1)	5 mg
Riboflavine (vitamin B2)	5 mg	Riboflavine (vitamin B2)	5 mg
Pyridoxine (vitamin B6)	5 mg	Pyridoxine (vitamin B6)	5 mg
Cyanocobalamin (vitamin B12)	10 mcg	Cyanocobalamin (vitamin B12)	10 mcg

- = None, Equiv. = Equivalent.

**Table 2 nutrients-14-00452-t002:** General characteristic of subjects according to supplementation group.

General Characteristics	Ferrous Bisglycinate Plus Folinic Acid			Control			*P1*	*P2*	*P3*
	Baseline	3 Months	6 Months	Baseline	3 Months	6 Months			
Age (year)	27.80 ± 3.66	-	-	29.11 ± 3.41	-	-	0.057	-	-
Weight (kg)	51.74 ± 4.77	57.08 ± 5.07	64.43 ± 5.44	49.04 ± 5.15	55.63 ± 5.57	61.90 ± 6.01	0.060	0.160	0.052
Body mass index (kg/m^2^)	20.81 ± 1.62	22.96 ± 1.74	25.92 ± 1.94	19.65 ± 1.49	22.29 ± 1.59	24.81 ± 1.64	0.054	0.059	0.057
Blood pressure (mmHg)									
Systolic	116.36 ± 7.28	118.04 ± 6.00	119.02 ± 6.14	114.43 ± 7.40	115.11 ± 6.15	117.13 ± 5.42	0.175	0.074	0.094
Diastolic	72.84 ± 6.48	73.65 ± 5.65	73.62 ± 5.73	73.96 ± 6.22	74.94 ± 5.48	75.64 ± 5.08	0.360	0.232	0.055
Pulse rate (bpm)	90.763 ± 4.80	92.05 ± 3.76	93.27 ± 2.85	92.19 ± 4.59	93.04 ± 4.37	92.87 ± 2.69	0.118	0.212	0.450
hs-CRP (mg/L)	0.91 ± 0.33	0.88 ± 0.28	0.87 ± 0.23	0.80 ± 0.28	0.85 ± 0.24	0.81 ± 0.22	0.054	0.583	0.208

Values are means ± SD. *P1* = Comparison of means between the two groups at baseline; *P2* = Comparison of means between the two groups at 3 months after supplementation; *P3* = Comparison of means between the two groups at 6 months after supplementation; significant differences at *p* < 0.05. - = None, hs-CRP = High sensitivity C-reactive protein.

**Table 3 nutrients-14-00452-t003:** Comparison of iron absorption at fasting, 1 h, and 2 h between ferrous bisglycinate plus folinic acid and control groups.

Time	Serum Iron (µg/dL)		*P*
	Ferrous Bisglycinate Plus Folinic Acid	Control	
Fasting	35.13 ± 5.35	34.85 ± 4.33	0.762
1 h	104.63 ± 11.86	88.16 ± 6.64	<0.001
2 h	142.07 ± 8.78	120.76 ± 9.70	<0.001

Values are means ± SD. *P* = Comparison of mean between the two groups; significant differences at *p* < 0.05.

**Table 4 nutrients-14-00452-t004:** Comparison of hematological and iron status at baseline and during the follow-up by supplementation.

Iron Status	Baseline			3 Months after Supplementation			6 Months after Supplementation		
	Ferrous Bisglycinate Plus Folinic Acid	Control	*P*	Ferrous Bisglycinate Plus Folinic Acid	Control	*P*	Ferrous Bisglycinate Plus Folinic Acid	Control	*P*
Hb (g/dL)	10.04 ± 0.83	10.17 ± 0.77	0.415	12.40 ± 0.68	11.78 ± 0.72	<0.001	12.82 ± 0.66	12.09 ± 0.60	<0.001
Mean Change				2.356 ± 0.69	1.61 ± 0.838	<0.001	2.78 ± 0.822	1.92 ± 0.89	<0.001
Erythrocytes (×10^12^/L)	2.23 ± 0.42	2.11 ± 0.30	0.066	3.93 ± 0.437	3.23 ± 0.36	<0.001	4.22 ± 0.30	3.41 ± 0.317	<0.001
Mean Change				1.69 ± 0.581	1.12 ± 0.33	<0.001	1.98 ± 0.49	1.30 ± 0.35	<0.001
Reticulocytes (×10^9^/L)	45.98 ± 4.53	43.15 ± 4.15	0.051	63.71 ± 6.69	55.77 ± 3.88	<0.001	69.91 ± 6.32	57.70 ± 4.48	<0.001
Mean Change				17.73 ± 7.99	12.62 ± 4.03	<0.001	23.93 ± 8.12	14.55 ± 5.02	<0.001
MCV (fL)	72.76 ± 3.77	70.38 ± 3.67	0.061	79.55 ± 3.70	75.81 ± 3.01	<0.001	81.75 ± 3.37	78.91 ± 3.38	<0.001
Mean Change				6.78 ± 4.45	5.43 ± 3.00	0.067	8.98 ± 4.72	8.53 ± 4.24	0.601
MCH (pg)	26.84 ± 3.19	25.59 ± 2.54	0.059	33.73 ± 1.83	30.06 ± 2.04	<0.001	34.64 ± 1.52	31.35 ± 2.00	<0.001
Mean Change				4.8 ± 2.489	3.905 ± 1.757	0.033	5.909 ± 2.619	4.981 ± 2.240	0.051
MCHC (g/dL)	26.84 ± 3.19	25.59 ± 2.54	0.059	33.73 ± 1.83	30.06 ± 2.04	<0.001	34.64 ± 1.52	31.35 ± 2.00	<0.001
Mean Change				6.89 ± 3.51	4.46 ± 1.86	<0.001	7.81 ± 3.49	5.76 ± 2.98	0.002
Transferrin Saturation (%)	19.90 ± 3.62	20.68 ± 2.88	0.1598	34.78 ± 3.92	28.96 ± 1.92	<0.001	36.47 ± 3.59	30.08 ± 2.11	<0.001
Mean Change				14.88 ± 3.62	8.28 ± 3.457	<0.001	16.58 ± 3.76	9.40 ± 3.43	<0.001
Ferritin (µg/L)	25.63 ± 3.128	23.90 ± 3.31	0.060	38.70 ± 4.04	30.12 ± 2.91	<0.001	40.45 ± 3.13	31.02 ± 1.70	<0.001
Mean Change				13.077 ± 2.886	6.218 ± 1.86	<0.001	14.78 ± 3.305	7.119 ± 3.18	<0.001

Values are means ± SD. *P* = Comparison of means between the two groups; significant differences at *p* < 0.05. Hb = Hemoglobin, MCV = Mean corpuscular volume, MCH = Mean corpuscular hemoglobin, MCHC = Mean corpuscular hemoglobin concentration.

**Table 5 nutrients-14-00452-t005:** Side effects between ferrous bisglycinate plus folinic acid and control groups.

Side Effects	Ferrous Bisglycinate Plus Folinic Acid *n* (%)	Control *n* (%)	*p*
Nausea			
- No	51 (92.73)	28 (52.83)	<0.001
- Yes	4 (7.27)	25 (47.17)	
Vomiting			
- No	54 (98.18)	47 (88.68)	0.058
- Yes	1 (1.82)	6 (11.32)	
Abdominal pain			
- No	53 (96.36)	32 (60.38)	<0.001
- Yes	2 (3.64)	21 (39.62)	
Bloating			
- No	53 (96.36)	34 (64.15)	<0.001
- Yes	2 (3.64)	19 (35.85)	
Constipation			
- No	51 (92.73)	17 (32.08)	<0.001
- Yes	4 (7.27)	36 (67.92)	
Diarrhea			
- No	54 (98.18)	50 (94.34)	0.359
- Yes	1 (1.82)	3 (5.66)	
Metallic taste			
- No	53 (96.36)	29 (54.72)	<0.001
- Yes	2 (3.64)	24 (45.28)	

**Table 6 nutrients-14-00452-t006:** Comparison of quality of life at baseline and during the follow-up by supplementation.

Quality of Life	Ferrous Bisglycinate Plus Folinic Acid			Control			*P1*	*P2*	*P3*
	Baseline	3 Months	6 Months	Baseline	3 Months	6 Months			
QOL-score	3.13 ± 0.45 ^a^	3.03 ± 0.44 ^b^	2.93 ± 0.47 ^c^	2.96 ± 0.42 ^a^	2.83 ± 0.38 ^b^	2.69 ± 0.38 ^c^	0.039	0.013	0.004

Values are means ± SD. Means in a row with superscript letters without a common letter differ within group; significant differences at *p* < 0.05. *P1* = Comparison of mean between the two groups at baseline; *P2* = Comparison of means between the two groups at 3 months after supplementation; *P3* = Comparison of means between the two groups at 6 months after supplementation; significant differences at *p* < 0.05.

**Table 7 nutrients-14-00452-t007:** Comparison of birth-weight between ferrous bisglycinate plus folinic acid and control groups.

	Iron Supplementation		*P*
	Ferrous Bisglycinate Plus Folinic Acid	Control	
Newborn weight (g)	3103.82 ± 270.85	2992.26 ± 254.86	0.029

Values are means ± SD. *P* = Comparison of means between the two groups; significant differences at *p* < 0.05.

## Data Availability

The datasets used and/or analyzed during the current study are available from the corresponding author upon reasonable request.

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
