# Peer review of "Efficacy and Safety of Ferrous Bisglycinate and Folinic Acid in the Control of Iron Deficiency in Pregnant Women: A Randomized, Controlled Trial"

_nutrients, 2022, doi:10.3390/nu14030452_

Round 1

Reviewer 1 Report

Reviewer Recommendation and Comments for:

Journal                        Nutrients (ISSN 2072-6643)

Manuscript ID             nutrients-1519764

Title                             The Efficacy and Safety of Ferrous Bisglycinate and Folinic Acid in the Control of Iron Deficiency in Pregnant Women: A Randomized Controlled Trial

This RCT of two different forms of iron and additional micronutrient supplementation during pregnancy describes implementation of a trial and results with public health implications. The trial appears to have been well-designed and adequately powered with 120 women randomized to one of two groups, however the description of the methods, the results and conclusions are lacking sufficient detail.

Major comments:

Throughout the paper the description of the differences between the “treatment” and “control” arms emphasizes the iron content of the supplement, without sufficient emphasis on the difference in folinic acid.  It would be helpful to reframe the whole paper as a comparison of the TWO differences in supplementation between groups.  There are TWO differences in micronutrients—iron and folic acid that are both in forms that are less commonly used among pregnant women.  Both of those factors should be clearly described in each section of the paper.

There is no mention of blinding in the study.  Were the participants blinded? Were the researchers? Were the physicians?  Either way, this needs to be described and potentially noted as a limitation in the discussion if there was no blinding.

There is no description of the source population.  Where were the pregnant women recruited? While there were no statistically significant differences in the maternal characteristics described between the treatment and control groups at baseline, there were a few differences that may have been clinically meaningful (i.e., body weight and CRP).  Sometimes, these types of differences show up because the clinical staff who are referring patients to the study have a conscious or unconscious bias as to which patients get referred to the study.  So that the reader can adequately assess these possibilities, there should be a detailed description of the source population and the referral process to the study.  Some questions that might be of interest to the reader include, Does the “Health Promotion Hospital” treat rural or urban women or both? Are high-risk women referred elsewhere? How prevalent is iron deficiency anemia in the source population? How long did it take to recruit 120 pregnant women with iron deficiency anemia?  What was the recruitment rate—did most women agree to participate or was it a selected-group of women who agreed?

There is little description of the implementation itself (including how long it took to recruit 120 women as noted above).  In the results, it is mentioned that 97% and 92% of participants took their supplements but the process for assessing supplementation was never described in the methods.  Did participants keep daily study records?  If so, how often were those reviewed with study staff, etc.?

This appears to be an important study with a potential impact that could be strengthened considerably with additional detail and reorganization and polish of the discussion to better highlight the key points.

Minor comments:

Abstract: What does “compliant” mean in the last sentence?  Perhaps a different word would be more appropriate.

Fig 1 needs editing.  It is blurry and it includes paragraph symbols.

Fig 2 needs obvious editing.

In the second paragraph of the discussion, line 209, “gave” looks like a typo that should be “treated.”

Reviewer 2 Report

The authors conduct a parallel-group randomized controlled trial (RCT) and aimed to examine the absorption, efficacy, and safety of oral ferrous bisglycinate plus folinic acid supplementation on hematological and iron status in pregnant women with iron deficiency. 

Comments:

  1. Abstract

The study design was a randomized controlled trial.

Blinded or Nonblinded Randomized Controlled Trials?

  1. Statistical analysis

For statistical analysis, it is intention-to-treat analysis or per-protocol analysis.

Data set includes all patients as originally allocated after randomization, treatment group (n=60) vs control group (n=60), or includes only those patients who completed the treatment originally allocated, treatment group (n=55) vs control group (n=53).

CMAJ. 2011 Apr 5; 183(6): 696.

Intention-to-treat analysis is a comparison of the treatment groups that includes all patients as originally allocated after randomization.

Per-protocol analysis is a comparison of treatment groups that includes only those patients who completed the treatment originally allocated.

  1. Statistical analysis in Tables 3, 4 6

Line 117, the repeated measures ANOVA with Bonferroni correction was performed.

Maybe, the authors should be used a mixed model or generalized model repeated measures analysis to assess the mean profile over time for each treatment group, such as provided 1)timecourse of mean change in serum iron from baseline by hour or change in serum iron from baseline to hour 2 (Table 3); 2)timecourse of mean change in iron status from baseline by month or change in iron status from baseline to month 6 (Table 4); 3)timecourse of mean change in quality of life from baseline by month or change in quality of life from baseline to month 6 (Table 6).

(Reference Lancet. 2014 May 10;383(9929):1637-1647.)

  1. Statistical analysis

Authors should be calculated the sample size of subjects per group as provided >80% power to detect such a difference.

Compute the sample size required to ensure high power when hypothesis testing.

Round 2

Reviewer 2 Report

No further comment